# Risk Stratification in Patients with Acute Pulmonary Embolism: Current Evidence and Perspectives

**DOI:** 10.3390/jcm11092533

**Published:** 2022-04-30

**Authors:** Antonio Leidi, Stijn Bex, Marc Righini, Amandine Berner, Olivier Grosgurin, Christophe Marti

**Affiliations:** 1Division of General Internal Medicine, Department of Medicine, Geneva University Hospitals, 1205 Geneva, Switzerland; antonio.leidi@hcuge.ch (A.L.); stijn.bex@hcuge.ch (S.B.); amandine.berner@hcuge.ch (A.B.); olivier.grosgurin@hcuge.ch (O.G.); 2Division of Angiology and Haemostasis, Department of Medicine, Geneva University Hospitals, 1205 Geneva, Switzerland; marc.righini@hcuge.ch

**Keywords:** risk assessment, pulmonary embolism, thrombolysis

## Abstract

Risk stratification is one of the cornerstones of the management of acute pulmonary embolism (PE) and determines the choice of both diagnostic and therapeutic strategies. The first step is the identification of patent circulatory failure, as it is associated with a high risk of immediate mortality and requires a rapid diagnosis and prompt reperfusion. The second step is the estimation of 30-day mortality based on clinical parameters (e.g., original and simplified version of the pulmonary embolism severity index): low-risk patients without right ventricular dysfunction are safely managed with ambulatory anticoagulation. The remaining group of hemodynamically stable patients, labeled intermediate-risk PE, requires hospital admission, even if most of them will heal without complications. In recent decades, efforts have been made to identify a subgroup of patients at an increased risk of adverse outcomes (intermediate-high-risk PE), who might benefit from a more aggressive approach, including reperfusion therapies and admission to a monitored unit. The cur-rent approach, combining markers of right ventricular dysfunction and myocardial injury, has an insufficient positive predictive value to guide primary thrombolysis. Sensitive markers of circulatory failure, such as plasma lactate, have shown interesting prognostic accuracy and may play a central role in the future. Furthermore, the improved security of reduced-dose thrombolysis may enlarge the indication of this treatment to selected intermediate–high-risk PE.

## 1. Introduction

Pulmonary embolism (PE) is the third most frequent cardiovascular disease and is associated with a high mortality burden, accounting for approximately 300,000 deaths in Europe every year [1,2]. PE is defined as the obstruction of a pulmonary artery, mostly resulting from the dislodgement of thrombotic material from the lower limbs. It has a wide variety of presentations, ranging from an asymptomatic incidental finding to circulatory collapse and sudden death. Diagnosis of PE often requires a sequential strategy combining a pre-test probability assessment, D-dimers measurement when indicated and thoracic imaging. Risk stratification immediately guides the management of acute PE, as it determines the need for urgent reperfusion therapy (high-risk PE) and identifies patients who can be safely treated as outpatients (low-risk PE). The remaining group of patients, called intermediate-risk PE, is highly heterogeneous with most of the patients recovering, but a significant proportion being at risk of complications. The present article will review the evidence supporting risk stratification and reperfusion strategies in the management of PE, with a particular focus on intermediate-risk pulmonary embolism.

## 2. Risk Stratification in Acute Pulmonary Embolism

Risk stratification is applied in various medical conditions to stratify patients’ severity in therapeutic trials or guide specific diagnostic or therapeutic interventions. Risk stratification often relies on prognostic scores built on clinical or biological parameters. Traditional steps in the development of a prognostic tool include derivation, internal and external validation and impact studies (i.e.**,** studies evaluating the benefit of a risk stratification strategy). While derivation and validation studies are plentiful, impact studies are scarce [3]. Risk stratification in the setting of acute PE includes three main steps: identification of patients at a high risk of early mortality, hence, requiring immediate reperfusion treatment; identification of patients at a low risk of complications who can be safely treated as outpatients; identification of patients with an increased risk of complications requiring hospitalization for close monitoring and potential primary or rescue reperfusion therapy (Figure 1) [4].

Nomenclature and definitions slightly differ between the European Society of Cardiology (ESC) and American Heart Association (AHA) guidelines; differences are highlighted in Table 1 [4,5]. This review mainly relies on the principles outlined in the 2019 ESC guidelines [4].

### 2.1. Step 1: Identification of High-Risk Patients

The first step in risk stratification of acute PE is the identification of patients at a high risk of early mortality. The most feared complication of acute PE is right ventricular overload and dysfunction which may lead to circulatory collapse and death. Therefore, patients with patent hemodynamic instability are considered as high-risk, according to the ESC criteria. Hemodynamic instability is defined by a systolic blood pressure (SBP) inferior to 90 mmHg for more than 15 min in the absence of hypovolemia, sepsis or arrhythmia; and/or the need of vasopressors in combination with end-organ hypoperfusion. In a recent systematic review including forty thousand patients with PE, 3.9% had high-risk PE. Short-term mortality was 19% among patients presenting with unstable PE versus 5.7% among patients with stable PE (OR 5.9; 95% CI 2.7 to 13.0) [6]. In a recent cohort of 7438 Chinese patients, the prevalence of unstable PE was 4.2% and mortality was 15.8% [7]. In high-risk acute PE, management relies on organ support and prompt reperfusion with thrombolytic therapies or percutaneous/surgical thrombectomy. The benefit of systemic thrombolysis has been demonstrated in small randomized controlled trials (RCTs) and large observational databases [6,8]. In the landmark study by Jerjes-Sanchez et al**.**, eight patients with high-risk PE were randomized to anticoagulation (AC) alone or in combination with streptokinase [8]. The four patients randomized to AC died and the four patients randomized to thrombolysis survived. However, this small size, open-label trial was limited by an imbalance between groups. Recent RCTs evaluating thrombolytic therapy usually excluded high-risk PE [9,10,11], while older studies did not separately report outcomes for high-risk PE [12,13]. In a large North American database including more than two million patients with PE, in-hospital mortality was 15% among patients with unstable PE receiving TT and 47% among patients with unstable PE treated with AC alone. However, 70% of unstable patients did not receive TT [14,15]. This underutilization may be explained by the reluctance of physicians to administer TT because of its bleeding complications. Moreover, the definition of high-risk PE in previous ESC criteria, based exclusively on the presence of hypotension, was probably too simplistic as systolic blood pressure should probably be considered as a continuous risk marker rather than a dichotomized variable [16]. In the 2019 recommendations, they have been enriched by evidence of end-organ dysfunction and the exclusion of alternative contributors of shock [4,17].

In summary, identification of high-risk patients based on the presence of hemodynamic instability is recommended for rapid diagnosis and prompt reperfusion therapy.

### 2.2. Step 2: Outpatient Management of Low-Risk Pulmonary Embolism

Historically, all patients with acute PE were admitted to the hospital. Short-term mortality prediction rules have, therefore, been developed to identify patients at a low risk of mortality who could be treated as outpatients. The pulmonary embolism severity index (PESI) was derived and internally validated in 2005 in a cohort of 15,531 patients. The PESI score comprises 11 clinical variables and stratifies patients into five severity classes [18]. A simplified version of the PESI (sPESI) was derived including six clinical variables, each scoring one point (Appendix A) [19]. According to a 2012 meta-analysis including 50,021 patients, the area under the curve (AUC) of sPESI was 0.79 for all-cause mortality with a pooled sensitivity of 0.92 and a pooled specificity of 0.38, which is similar to the original PESI score. The pooled mortality was 2% among patients with PESI class I or II and 1.8% among patients with 0 points in sPESI (Table 2) [20]. A non-inferiority interventional study compared PESI-based outpatient management with hospital admission [21]. Three hundred and forty-four patients with low-risk PE (PESI class I-II) were randomly allocated to outpatient versus inpatient management. Ninety-day mortality was non-inferior in outpatients compared to patients admitted in-hospital (0.6% in both groups, upper confidence limit (UCL) for difference 2.1%) as well as PE recurrence (0.6% in outpatients versus 0% for inpatients, UCL 2.7%). At three months, three outpatients (1.8%), but no inpatients, developed major bleeding (UCL 4.5%).

Interestingly, when a low-risk PESI is combined with the absence of right ventricular dysfunction (RVD), 30-day mortality decreases even further (0.2–0.3%) [23,24]. In a single arm study evaluating the early discharge of low-risk PE (normotensive, absence of RVD and absence of serious comorbidities) treated with rivaroxaban, the rate of major bleeding at 3 months was low (1.2%) [25]. However, this increased sensitivity is obtained at the expense of a significantly lower proportion of patients identified as low risk [26]. An alternative approach to identify low-risk patients is the use of the Hestia criteria (Appendix A), consisting of a checklist of eleven criteria requiring hospital admission. Home treatment of patients without these criteria has been shown to be safe and non-inferior to sPESI-based home treatment [27,28,29].

In summary, outpatient treatment appears to be safe for low-risk PE patients identified by PESI, sPESI or Hestia criteria and absence of RVD.

## 3. Step 3: Further Classification of Intermediate-Risk Pulmonary Embolism

About 4% and 40% of acute pulmonary embolisms are categorized as high risk and low risk, respectively [30]. The remaining patients (i.e.**,** normotensive patients with PESI III-V or sPESI ≥ 1) are classified as intermediate-risk PE with an overall 30-day mortality between 5% and 15% [18,19,31]. This group of patients is highly heterogeneous, with the vast majority experiencing a favorable outcome with AC alone, and a small, albeit significant, proportion requiring rescue reperfusion and cardiopulmonary resuscitation. For this reason, efforts have been made in recent decades to identify a subgroup of patients at risk of deterioration who might benefit from initial aggressive therapy and/or admission to a monitored unit. Clinical scores, biological and radiological markers of right ventricular overload and circulatory failure, alone or in combination, have been proposed to further stratify intermediate-risk acute PE.

### 3.1. Clinical Scores

Despite a high sensitivity and negative predictive value, PESI and sPESI lack specificity to predict early mortality (Table 2). Moreover, these scores rely heavily on demographic and co-morbid conditions rather than the severity of the acute PE event. PE-attributable mortality represents less than half of the overall 3-month mortality among patients with an acute PE [32,33]. Clinical scores alone are, therefore, inadequate to guide admission to monitored units or to initiate reperfusion therapies. Right ventricular dysfunction with circulatory collapse is the most common mechanism leading to fatal PE. Various markers of RVD and circulatory failure have been investigated as potential tools to further stratify normotensive patients; they are detailed in the following sections and summarized in Table 3.

### 3.2. Markers of Right Ventricular Dysfunction


*Cardiac troponin*


The prognostic value of Troponin I and T has been evaluated in a meta-analysis including 1985 patients [36]. Elevated troponin was associated with increased short-term mortality in the whole cohort and in the subgroup of normotensive patients (OR 5.90, 95% CI 2.68 to 12.95, overall short-term mortality 17.9%). Subsequent meta-analyses questioned the prognostic value of elevated troponin in normotensive patients (positive likelihood ratio 2.13, negative likelihood ratio 0.51) and it has been suggested to combine it with other prognostic factors [34,37].


*Brain natriuretic peptides*


Brain natriuretic peptide (BNP) and its N-terminal portion (NT-proBNP) are secreted by cardiomyocytes in response to ventricular stretching due to volume or pressure overload. The prognostic value of natriuretic peptides has been evaluated in at least eight studies [35,38]. In a meta-analysis, the pooled relative risk for 30-day mortality was 9.5 (95% CI 3.1 to 28.6) for BNP and 8.3 (95% CI 3.6 to 19.3) for NT-proBNP [35]. A meta-analysis of five more recent studies reported a relative risk for the complicated clinical course of 5.63 (95% CI 2.77 to 11.43) when the NT-proBNP value was over 1000 pg/mL [39]. Interestingly, troponin and brain natriuretic peptides seem to have an additive prognostic value [38,40,41].

Other biomarkers, such as serum creatinine, heart fatty acid-binding protein and copeptin, have been studied but are less extensively validated and/or not available in clinical practice [42,43,44,45].


*Computer tomography pulmonary angiography*


CTPA signs of RVD include an elevated RV/left ventricular (LV) end diastolic diameter ratio (cut-off of 0.9 or 1.0), interventricular septum bowing, pulmonary artery enlargement, and retrograde reflux of contrast into the vena cava [46]. The right-to-left ventricular ratio can be easily estimated using the largest transverse diameters which may be measured on different CTPA slices. Septum bowing has an excellent specificity (98%) but poor sensitivity (31%), and inter-observer reproducibility limits its clinical utilization [46,47]. CT obstruction indexes have also been proposed by Qanadli et al. and [48] have shown to be associated with increased mortality, mostly among patients without comorbidities [49].


*Bedside echocardiography*


Increased right-to-left ventricular ratio, hypokinesis of the free RV wall and the presence of pulmonary hypertension estimated from tricuspid regurgitation velocity have been reported to be associated with an increased risk of early complications. More advanced measures, such as the ratio of tricuspid annular plane systolic excursion to pulmonary arterial systolic pressure (TAPSE/PASP) are being investigated to stratify the risk among PE patients, but they are impractical for daily use and bedside stratification [50]. More recently, additional echocardiographic markers such as left or right ventricular outflow tract velocity time integral (LVOT and RVOT VTI) and stroke volume index have been reported to be associated with death or clinical deterioration, with interesting discriminative performance among intermediate-risk patients [51,52,53,54].

Despite a consistent association with short-term mortality, markers of RVD have poor diagnostic performances when they are used as a stand-alone test (Table 3) [17,35]. They have, therefore, been combined in current risk stratification guidelines.

### 3.3. Current Stratification of Intermediate-Risk Pulmonary Embolism

The ESC 2019 risk stratification of patients with acute PE relies on the three-step process described above, based on the presence of hemodynamic instability, clinical scores (PESI or sPESI) and the combined presence of two markers of RVD (Figure 1 and Table 1). This revision of the previously published 2014 criteria allows us to identify a subgroup of intermediate-risk PE patients at risk of short-term circulatory collapse or mortality, labelled as intermediate–high risk. These criteria were conceived following expert opinions, and no impact study has been published to date. In a prospective cohort of 1015 patients with normotensive acute PE, the ESC 2019 criteria classified 347 (34%), 571 (56%) and 97 (9.6%) of patients as low, intermediate–low and intermediate–high-risk PE, respectively. All cause 30-day mortality was significantly higher in intermediate–high-risk patients (10%) than in those with low risk or intermediate–low risk (4%) [55].

## 4. Reperfusion Therapy for Intermediate–High-Risk Pulmonary Embolism

Systemic thrombolysis using plasminogen activators is the most widely studied reperfusion strategy. Tissue plasminogen activators (tPA), such as urokinase, alteplase or tenecteplase, have a fibrinolytic effect, allowing clot dissolution and improvement of hemodynamic parameters in patients with high-risk PE [8,13]. Several randomized controlled trials aimed to evaluate their potential benefit among normotensive patients with elevated markers of RVD. The most informative evidence is provided by the European Pulmonary Embolism Trombolysis (PEITHO) trial [10]. This large RCT included 1005 patients with acute PE and RVD on imaging (CTPA or echocardiography) and myocardial injury (elevated troponin T or I), corresponding to the current intermediate–high-risk category. Patients were randomly assigned to unfractionated heparin (UFH) plus tenecteplase or UFH alone. The incidence of the primary outcome (death or hemodynamic collapse within 1 week) was significantly lower among patients allocated to tenecteplase than to UFH alone (2.6% vs. 5.6%, *p* = 0.02). This difference was mainly driven by an increased risk in hemodynamic decompensation among patients allocated to UFH (5.0% vs. 1.6%) while mortality was low and did not significantly differ (1.2% vs. 1.8%). This was counterbalanced by a significant increase in the risk of both major (11.5% vs. 2.4%) and intracranial bleeding (2.0% vs. 0.2%). When thrombolysis studies exclusively including acute PE with RVD are pooled, the uncertain benefit in overall mortality is mitigated by the significant increase in both major and intracranial bleeding [11]. Of note, the increased risk of major bleeding was more pronounced in studies using tenecteplase than in those using alteplase, but direct comparison studies are lacking to confirm this observation [56].

## 5. Toward a Better Identification of Thrombolysis Candidates among Intermediate-Risk Pulmonary Embolism

The current ESC classification seems to have an insufficient positive predictive value to identify a subgroup of intermediate-risk patients warranting more aggressive therapy. The benefits of a full-dose systemic thrombolysis are outweighed by bleeding risks, particularly when using tenecteplase. Two strategies have been identified by researchers to enhance the benefit-risk ratio: identifying patients at a higher basal risk and improving the safety of TT.

### Identifying Patients at Higher Basal Risk: Markers of Circulatory Failure and Alternative Scores

Various alternative prediction rules, including early markers of circulatory failure, have been studied to identify normotensive patients with a higher risk of hemodynamic collapse.


*Plasma lactate*


Plasma lactate is an important prognostic marker of organ dysfunction and is widely used in patients with sepsis or trauma [57]. Several studies evaluated the prognostic value of plasma lactate among patients with acute PE [58,59]. A retrospective study including 287 patients with acute PE reported a significant association between plasma lactate levels above 2 mmol/L and in-hospital mortality (OR 4.6; 95% CI 1.57 to 13.53) [58]. An association with 30-day mortality was subsequently observed in a prospective study (HR 11.67; 95% CI 3.32 to 41.03) [59]. Interestingly, this association was independent of shock state, hypotension, RVD or elevated troponin. A single center registry of 419 consecutive PE patients confirmed the association of elevated venous lactate with adverse outcomes, and found that levels above 3.3 mmol/L had the best predictive performance for in-hospital adverse events (PPV 0.27 and NPV 0.97) [60]. Moreover, adding venous lactate levels to the ESC 2019 risk criteria allowed us to further fine-tune stratification. Intermediate–high-risk patients with venous lactate ≥3.3 mmol/L had a 27.5% prevalence of adverse events, versus 6.8% if lactate was <3.3 mmol/L. Intermediate low-risk patients with lactate levels <2.3 mmol/L were at a low risk of adverse events (0.6%, versus 12.2% if ≥2.3 mmol/L) [60].


*BOVA score*


The Bova score was derived from pooled results of six prospective studies, including 2874 patients with hemodynamically stable acute PE [61]. Model predictors included heart rate, SBP, biomarkers (cardiac troponin or BNP) and echocardiography (Table 4 and Appendix A) [61]. RVD was defined by the presence of RV/LV >0.9 or 1, RV free wall hypokinesis, RV end-diastolic diameter >30 mm or estimated systolic pulmonary artery pressure > 30 mmH. The primary composite outcome was PE-related death, hemodynamic collapse or recurrent PE at 30 days. Thirty-day complications differed significantly across categories of the model (0–2 points 4.2%; 3–4 points 10.8%; >4 points 29.2%). The area under the ROC curve was 0.73 (95% CI 0.68–0.77) and 5.8% of patients were classified in the stage III category. Recently, a meta-analysis including the derivation study and eight prospective and retrospective external validation cohort studies were conducted [62]. The pooled cumulative incidence of PE-related complications (PE-related death, hemodynamic collapse or recurrent PE) at 30 days was 3.8% for stage I, 10.8% for stage II and 19.9% for stage III (1.9, 5.5 and 12.1 for 30-day PE mortality) with an AUC of 0.73 (Table 4) [63]. In another retrospective cohort including 994 normotensive patients, 5.9% of patients were classified in the stage III category. Death or hemodynamic collapse at 7 days occurred in 18.6% of patients in the stage III category. When lactate elevation was incorporated into an extended Bova score, the proportion of patients in the stage III category increased to 11.2 %, with a primary outcome rate of 25.9 %. Hemodynamic collapse by day 7 occurred in 15.3% of patients in the class III category according to the standard BOVA score, compared to 24.1% in the model including lactate elevation [64].


*TELOS score*


The TELOS score was derived in a prospective cohort of 496 normotensive PE patients. The primary outcome was PE-related death or hemodynamic collapse within 7 days. A model including RVD, troponin and plasma lactate elevation resulted in a 17.9% PPV [65]. The TELOS rule was further validated by the same group in a prospective cohort of 994 normotensive patients. A total of 5.9% of patients were allocated to the intermediate–high-risk category according to the TELOS criteria, with a cumulative incidence of the primary outcome (death or hemodynamic collapse at 7 days) of 21.1% (Table 5) [64].


*SHIELD score*


The SHIELD score was derived from a retrospective monocentric cohort of 554 normotensive patients and was externally validated. Predictors of the model included shock index ≥1, hypoxemia, lactate elevation and signs of RVD (i.e., elevated troponin, NT-pro BNP and RV/LV ratio >1 using CTPA) [68]. The risk of 30-day mortality or rescue thrombolysis for each tercile was 0.6%, 1.8% and 16.4% (AUC 0.90, 95% CI 0.85 to 0.94) in the derivation cohort and 0.6%, 1.9% and 15.3% (AUC 0.82; 95% CI 0.75 to 0.87) in the external validation cohort.


*Other scores*


Lankeit et al. derived a clinical prediction rule including heart fatty acid-binding protein (H-FABP), syncope and heart rate (FAST score). The positive predictive value was 20.5% and the AUC was 0.85 (95% CI 0.75 to 0.95) [69]. This score was validated in another cohort of the same center with a positive predictive value of 18.9% and an AUC of 0.82 (95% CI 0.75 to 0.89) [66].

A prospective study including 268 normotensive PE patients showed an association between copeptin level >24 pmol/L and 30-day mortality or adverse outcome. The positive predictive value was 11% (95% CI 7 to 19%) and the negative predictive value was 98% (95% CI 95 to 99%), suggesting that this biomarker could be combined with markers of RVD such as NT-proBNP or highly sensitive troponin T [42].

Other scores combining clinical variables, imaging and biomarkers have been studied with less extensive validation [70,71].


*Between score comparison*


Vanni et al. compared the prognostic accuracy of the ESC 2014 criteria, TELOS and Bova score in a cohort of 994 normotensive patients with PE. The Bova and TELOS scores classified the same proportion of patients in the intermediate–high-risk category (5.9 and 5.7%) with a similar rate of early adverse events (18.6 and 21.1% 7-day death or hemodynamic collapse), while the ESC criteria classified a higher proportion of patients in the intermediate–high-risk category (12.5%, *p* < 0.001) with a lower rate of events (13% *p* = 0.18) [54]. Diagnostic performances of several existing scores are summarized in Table 5. While the risk of adverse events was comprised between 7% and 15% in the intermediate–high-risk group according to the ESC criteria, recent prediction rules, including markers of circulatory dysfunction (plasma lactates) or adjunction of plasma lactates to the Bova or ESC 2019 criteria, appeared to have a promising positive predictive value with event rates of around 25%. However, some variation in their PPVs was observed across studies, which is partly explained by variations in the outcome definitions. Moreover, the plasma lactate cut-off varied across studies and the optimal cut-off remains to be determined.

## 6. Expected Benefits from a Better Identification of Intermediate–High-Risk Patients

As discussed above, recent prediction rules, including markers of circulatory dysfunction, might allow us to identify a subgroup of patients with a significantly increased (>25%) risk in adverse events who might benefit from a more aggressive therapy. Figure 2 illustrates the extrapolated benefits of thrombolytic therapy in a theoretical cohort of patients with a 17.1% basal risk of adverse outcomes. This basal risk was obtained by combining the basal risk observed in the PEITHO trial and the positive likelihood ratio of elevated blood lactates [9]. Net benefits and harms were computed based on the relative risks reported in our previous meta-analysis on thrombolytic therapy in PE [11]. These extrapolations are mainly illustrative and should be evaluated in interventional studies, as bleeding complication risks might also increase among patients with a higher basal risk of a PE-related adverse event.

## 7. Improving the Safety Profile: Reduced-Dose Thrombolytic Therapy

Another strategy to optimize the risk-benefit ratio of TT among intermediate–high-risk PE is the reduction in TT bleeding complications. In this perspective, several trials investigated reduced-dose systemic thrombolysis regimens. In the Moderate Pulmonary Embolism Treated with Thrombolysis (MOPETT) trial, 121 patients with acute PE and a high thrombotic burden were randomized to half-dose tPA and heparin versus heparin alone. Half-dose thrombolysis was associated with a lower rate of pulmonary hypertension or recurrent PE at 28 months (16% versus 63% in the AC group) and no significantly increased risk of bleeding [72]. Similarly, a systematic review including 5 small-size randomized trials suggested a similar efficacy and reduced bleeding complications when half-dose was compared to standard dose thrombolysis [73]. These preliminary results based on a limited sample of patients need to be further confirmed.

Catheter-based thrombolytic therapies, including catheter-released thrombolysis (CRT), ultrasound-assisted thrombolysis and mechanical fragmentation or aspiration, alone or in combination, are alternative strategies to reduce bleeding risks associated with systemic TT [74,75]. Catheter-based therapies are an evolving technology and several devices have been shown to improve echocardiographic signs of RVD in single-arm studies [75,76]. Their impact regarding clinically relevant outcomes should be further evaluated in randomized controlled studies [77]. These alternatives may be proposed to high-risk patients at an increased risk of bleeding. More recently, circulatory support using extracorporeal membrane oxygenation (ECMO) in combination with surgical embolectomy has been proposed for high-risk patients with high in-hospital survival rates [78].

## 8. Evidence to Come: The PEITHO-3 Study

The ongoing PEITHO-3 study is expected to add further evidence for the management of intermediate–high-risk PE. This ongoing, multicentric, randomized controlled trial aims to combine the two previously discussed strategies. Identification of patients with a higher basal risk of adverse events will be obtained based on a retrospective analysis of the PEITHO population by adding clinical markers of severity (SBP ≤ 110 mmHg, respiratory rate >20/min, history of chronic heart failure) to the ESC intermediate–high-risk criteria with an expected event rate of 11.2% [79,80]. At the same time, a reduced dose of alteplase (0.6 mg/kg) will be used. The PEITHO-3 trial plans to include 650 patients with, expected to be completed in September 2025 [80]. The expected benefits in the hypothesized PEITHO-3 population are illustrated in Figure 3, assuming a constant relative effect of treatment and a constant basal bleeding risk with important uncertainty regarding the increase in bleeding complications with reduced-dose TT.

## 9. Conclusions

The management of patients with acute PE requires an accurate step-by-step risk stratification. Hemodynamic instability allows to quickly detect high-risk patients who will benefit from TT; clinical prediction rules, such as PESI or sPESI, allow us to identify low-risk patients who can be safely treated as outpatients. The intermediate-risk group is composed of a highly heterogeneous group of patients, most of whom will experience favorable short-term outcomes. Individual evaluations by dedicated multidisciplinary PE response teams may be useful to decide the optimal treatment strategy [81,82]. Several prediction models combining clinical variables, biomarkers and medical imaging have been developed to identify the subgroup of patients with a significant risk of early adverse events who might benefit from early aggressive management. The inclusion of plasma lactate may increase the performance of risk stratification models, but it needs to be further validated in future studies. Optimization of reperfusion strategies by using reduced regimens of thrombolysis or catheter-directed reperfusion techniques might contribute to improving the prognosis of patients by limiting the side effects. Results of ongoing studies (NCT04430569) will clarify the benefits of initial reperfusion strategies in this subgroup of patients. While waiting for these results, current guidelines recommend admitting patients of the intermediate–high-risk category to a monitored unit for early detection of clinical deterioration and the potential need for rescue reperfusion.

## Figures and Tables

**Figure 1 jcm-11-02533-f001:**
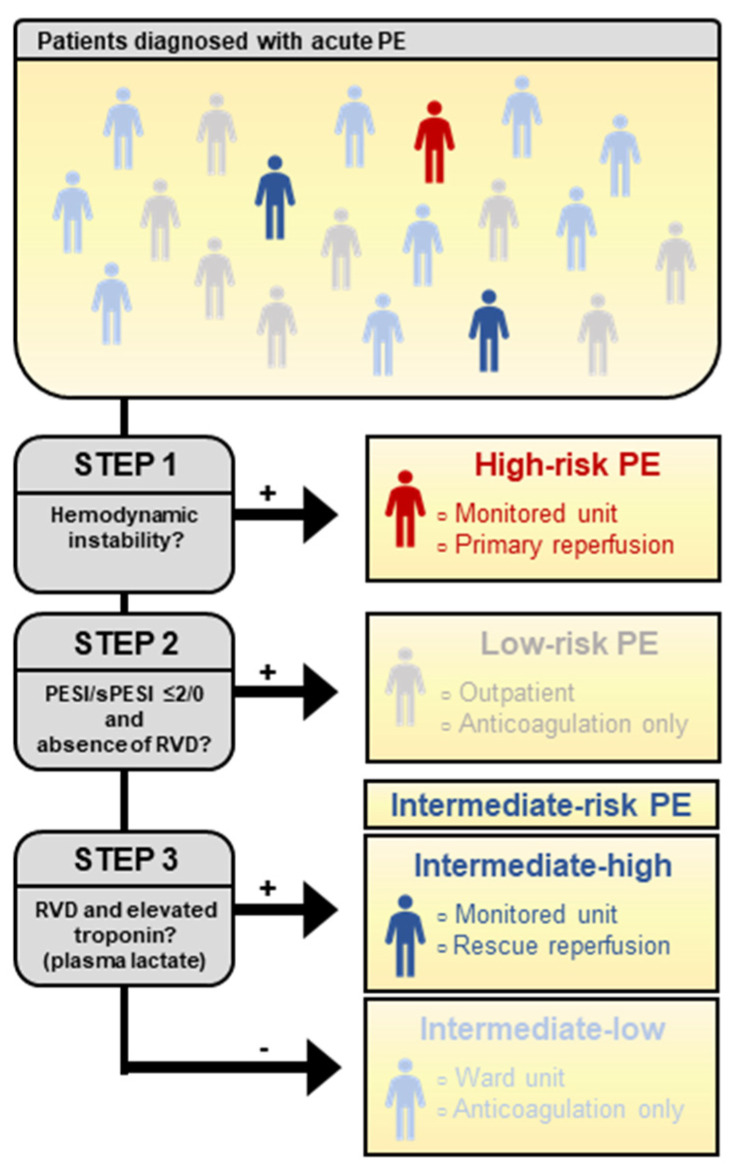
Step-by-step risk stratification in acute pulmonary embolism.

**Figure 2 jcm-11-02533-f002:**
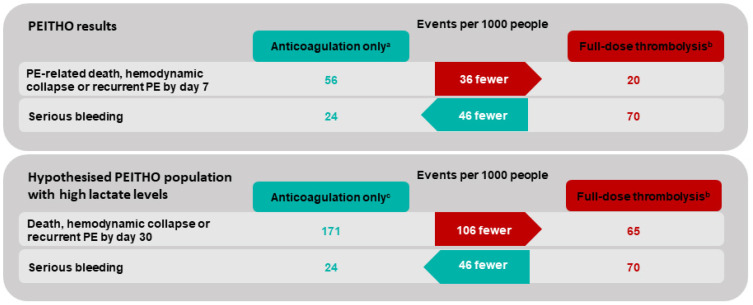
Extrapolated effect of full-dose thrombolysis in cohorts of patients with a different basal risk of an early adverse event. ^a^ Basal risk according to PEITHO trial [10], ^b^ relative effect according to Marti et al. [11]. ^c^ Basal risk obtained by combining the risk of the PEITHO population and the prognostic modulation obtained by adding lactate [60].

**Figure 3 jcm-11-02533-f003:**
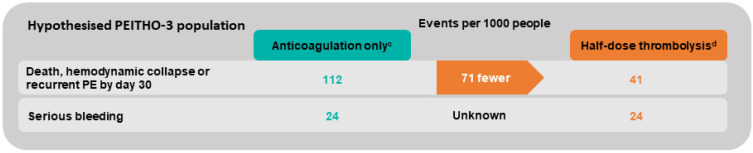
Extrapolated effect of half-dose thrombolysis in the hypothesized PEITHO-3 population. ^c^ Basal risk derived from applying the PEITHO-3 inclusion criteria and outcome to the PEITHO population [79]. ^d^ Relative treatment effect of full-dose thrombolysis is applied, and no increased risk of serious bleeding is assumed, according to a meta-analysis on half-dose thrombolysis [73].

**Table 1 jcm-11-02533-t001:** Nomenclature in current European and American guidelines.

Nomenclature	Hemodynamic Instability	RVD	Elevated Troponin	PESI > Class II or sPESI > 0
**European Society of Cardiology (ESC) 2019**				
High risk	+	(+)	(+)	+
Intermediate–high risk	−	+	+	+ *
Intermediate–low risk	−	One or none		+ *
Low risk	−	−	(−)	−
**American Heart Association (AHA) 2011**				
Massive	+	(+)	(+)	NA
Submassive	−	One or both		NA
Low risk	−	−	−	NA

RVD: right ventricular dysfunction; PESI: pulmonary embolism severity index; sPESI: simplified pulmonary embolism severity index; NA: not assessed. * Presence of RVD despite PESI ≤ 2 or sPESI 0 classifies patients in intermediate-risk category.

**Table 2 jcm-11-02533-t002:** Operative characteristics of original and simplified pulmonary embolism severity index for early all-cause mortality [22].

Prediction Index	Validation Cohorts (Patients)	Sensitivity (95% CI)	Specificity (95% CI)	PLR (95% CI)	NLR (95% CI)
**PESI**	19 (23,997)	0.89 (0.87–0.90)	0.49 (0.44–0.53)	1.72 (1.57–1.89)	0.22 (0.18–0.25)
**sPESI**	9 (26,610)	0.92 (0.89–0.94)	0.38 (0.32–0.44)	1.47 (1.28–1.68)	0.20 (0.13–0.31)

PESI: pulmonary embolism severity index; sPESI: simplified pulmonary embolism severity index; CI: confidence interval; PLR: positive likelihood ratio; NLR: negative likelihood ratio.

**Table 3 jcm-11-02533-t003:** Prognostic value of markers of right ventricular dysfunction for short-term mortality.

Marker	Sensitivity (95% CI)	Specificity (95% CI)	PLR (95% CI)	NLR (95% CI)
**Troponin** [34]	0.66 (0.61 to 0.70)	0.66 (0.65 to 0.67)	2.13 (1.84 to 2.47)	0.51 (0.40 to 0.60)
**BNP** [35]	0.88 (0.65 to 0.96)	0.70 (0.64 to 0.75)	2.13 (1.84 to 2.47)	0.51 (0.40 to 0.60)
**NT-proBNP** [35]	0.93 (0.14 to 1.00)	0.58 (0.14 to 0.92)	2.93 (2.28 to 3.77)	0.17 (0.05 to 0.58)
**RVD US** [35]	0.70 (0.46 to 0.86)	0.57 (0.47 to 0.66)	1.48 (1.05 to 2.08)	0.82 (0.65 to 1.03)
**RVD CT** [35]	0.65 (0.35 to 0.85)	0.56 (0.39 to 0.71)	1.63 (1.27 to 2.08)	0.53 (0.31 to 0.89)

CI: confidence interval; PLR: positive likelihood ratio; NLR: negative likelihood ratio; BNP: brain natriuretic peptide; NT-proBNP: N-terminal brain natriuretic peptide; RVD: right ventricular dysfunction; US: ultrasonography; CT: computer tomography.

**Table 4 jcm-11-02533-t004:** Components of the Bova score.

Predictor	Points
SBP 90–100 mmHg	2
Elevated troponin	2
RV dysfunction	2
Heart rate > 100/min	1

**Table 5 jcm-11-02533-t005:** Prognostic value of stratification scores dichotomized at an intermediate–high-risk level.

Score	Sensitivity	Specificity	PLR	PPV	Outcome
**Scores including Plasma Lactate**					
**ESC 2019 + lactate** [60]	0.33 (0.16 to 0.55)	0.95 (0.92 to 0.97)	6.27 (3.11 to 12.66)	**0.27**	In-hospital adverse outcome
**Bova + lactate** [64]	0.46 (0.34 to 0.58)	0.91 (0.90 to 0.92)	5.16 (3.55 to 7.13)	**0.26**	Adverse 7-day outcome
**TELOS** [64]	0.19 (0.11 to 0.30)	0.95 (0.95 to 0.96)	3.94 (2.04 to 7.15)	**0.21**	Adverse 7-day outcome
**Scores without plasma lactate**					
**Bova** [66]	0.48 (0.30 to 0.67)	0.86 (0.82 to 0.90)	3.41 (2.11 to 5.52)	**0.19**	Adverse 30-day outcome
**ESC 2014** [66]	0.80 (0.61 to 0.91)	0.69 (0.64 to 0.73)	2.60 (2.00 to 3.30)	**0.15**	Adverse 30-day outcome
**ESC 2019** [67]	0.52 (0.34 to 0.70)	0.79 (0.77 to 0.82)	2.5 (1.7 to 3.7)	**0.07**	In-hospital adverse outcome

ESC: European Society of Cardiology; PLR: positive likelihood ratio; PPV: positive predictive value.

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
