# Peer review of "Risk Stratification in Patients with Acute Pulmonary Embolism: Current Evidence and Perspectives"

_jcm, 2022, doi:10.3390/jcm11092533_

Round 1

Reviewer 1 Report

This is a comprehensive review of current strategies for risk stratification of patients presenting with acute PE. The authors should be congratulated on the enormous effort to compile best evidence and synchronize it in a cohesive, clinically relevant manner. 

The reviewer has several suggestions for the authors to consider. 

The authors explored in detail the need for risk stratification in normotensive patients with intermediate risk PE. There have been recent studies demonstrating important role of novel echocardiographic measures to further risk stratify patients with PE. Authors should consider including these markers (RVOT VTI, LVOT VTI, stroke volume index)and the relevant studies:

Brailovsky Y, Lakhter V, Weinberg I, et al. Right ventricular outflow doppler predicts low cardiac index in intermediate risk pulmonary embolism. Clin Appl Thromb. 2019

Yuriditsky E, Mitchell OJL, Sista AK, et al. Right ventricular stroke distance predicts death and clinical deterioration in patients with pulmonary embolism. Thromb Res. 2020;195:29–34.

Yuriditsky E, Mitchell OJL, Sibley RA, et al. Low left ventricular outflow tract velocity time integral is associated with poor outcomes in acute pulmonary embolism. Vasc Med (United Kingdom). 2020;25:133–40.

Prosperi-Porta G, Solverson K, Fine N, Humphreys CJ, Ferland A, Weatherald J. Echocardiography-derived stroke volume index is associated with adverse in hospital outcomes in intermediate-risk acute pulmonary embolism: a retrospective cohort study. Chest. 2020;158:1132–42.

As authors pointed out, hemodynamic instability is the main determinant of poor outcome. Clinical criteria often dichotomize patients based on “”hypotension” as defined by SBP<90 mmHg. Recent data have suggested that perhaps SBP is more of a continuous risk measure, rather than a dichotomized variable, although dichotomized variable is easier to apply clinically. Authors should consider mentioning this point and the associated study. 

Quezada, D. Jiménez, B. Bikdeli, et al., Systolic blood pressure and mortality in acute symptomatic pulmonary embolism, International Journal of Cardiology, 2019

Other points of clarification: 

Page 3

“In high-risk acute PE, manage-85 ment relies on organ support and prompt reperfusion with thrombolytic therapies.” Consider revising a sentence to include other “”advanced therapeutic options such as percutaneous or surgical embolectomy or mechanical circulatory support” Your sentence may be perceived to suggest that the only option available for high risk PE is thrombolytic therapy.

Page 5

“Despite a high sensitivity and negative predictive value, PESI and sPESI lack specificity 159 and their positive predictive value (PPV) for early mortality is low (Table 2).”

The table that you reference describes Sensitivity, specificity, PLR, and NLR, and while they are related to PPV, it is not the same, consider revising the sentence or the table to stay consistent. 

Page 10

“7. Improving the safety profile: reduced-dose thrombolytic therapy”

Consider adding percutaneous embolectomy devices which do not use any thrombolytic therapy and therefore may reduce the bleeding risk associated with the thrombolytic use.

Akhilesh K. Sista et al. J Am Coll Cardiol Intv 2021; 14:319-329.

Thomas Tu et al. J Am Coll Cardiol Intv 2019; 12:859-869.

Author Response

Answer to reviewer’s comments:

We would like to thank the reviewers and the Editor for the thoughtful comments and careful review of our manuscript that have allowed us to improve the quality of the reporting. Please find below a point-by-point response to the comments. Editor’s and Reviewers comments appear in italic and answers to Reviewers in black plain text.

Reviewers' comments:

Academic Editor: “I cordially thank the authors for the submission of the manuscript that covers the management of the life-threatening clinical condition, such as the acute pulmonary embolism.
I read the manuscript and found it interesting and useful for the clinical practice.”

  • We thank the Editor for this supporting comment

  1. I would like to kindly ask the authors to include more data from the review literature souces, such as:
    *Risk Stratification in Acute Pulmonary Embolism: Half of the Way There? – authors Myc et al.
    *Refining Risk Stratification in Nonmassive Acute Pulmonary Embolism – authors Kay et al.
    *Comparison of 4 Acute Pulmonary Embolism Mortality Risk Scores in Patients Evaluated by Pulmonary Embolism Response Teams – authors Barnes et al.
  • We thank the Editor for suggesting these relevant references which have been added to the appropriate sections (Outpatient treatment of low-risk patients and Computer tomography pulmonary angiography section).

*Comparison of 5 acute pulmonary embolism mortality risk scores in patients with COVID-19 – authors Rodrigues et al.

  • As early mortality is heavily dependent on the severity of pulmonary involvement among COVID-19 patients, we believe that this subgroup of patients differs from the general population of patients with acute PE and would therefore suggest not to address this specific population in the context of this review.

*Trends in risk stratification, in-hospital management and mortality of patients with acute pulmonary embolism: an analysis from the China pUlmonary thromboembolism REgistry Study (CURES)

  • We thank the reviewer for this relevant reference regarding a Chinese cohort of patients with acute PE. Data from this cohort has been added to the high-risk section

*Acute pulmonary embolism – author Howard L...
found in the PubMed database after submission of the key words „acute pulmonary embolism“ AND „risk stratification“

  • In order to limit the number of cited references, we would suggest not to mention this general narrative review about acute PE.
  1. I would like to kindly ask the authors to include more tables or schemes to manifest the risk stratification strategy
  • We thank the Editor for accepting inclusion of an additional Table in the manuscript. In addition to the 4 tables and 3 figures already included, an additional table detailing the components of the BOVA score has been added to the main manuscript.

Reviewer #1: “This is a comprehensive review of current strategies for risk stratification of patients presenting with acute PE. The authors should be congratulated on the enormous effort to compile best evidence and synchronize it in a cohesive, clinically relevant manner.”

  • We thank reviewer #1 for this supporting comment.

“The reviewer has several suggestions for the authors to consider.

The authors explored in detail the need for risk stratification in normotensive patients with intermediate risk PE. There have been recent studies demonstrating important role of novel echocardiographic measures to further risk stratify patients with PE. Authors should consider including these markers (RVOT VTI, LVOT VTI, stroke volume index) and the relevant studies:

Brailovsky Y, Lakhter V, Weinberg I, et al. Right ventricular outflow doppler predicts low cardiac index in intermediate risk pulmonary embolism. Clin Appl Thromb. 2019

Yuriditsky E, Mitchell OJL, Sista AK, et al. Right ventricular stroke distance predicts death and clinical deterioration in patients with pulmonary embolism. Thromb Res. 2020;195:29–34.

Yuriditsky E, Mitchell OJL, Sibley RA, et al. Low left ventricular outflow tract velocity time integral is associated with poor outcomes in acute pulmonary embolism. Vasc Med (United Kingdom). 2020;25:133–40.

Prosperi-Porta G, Solverson K, Fine N, Humphreys CJ, Ferland A, Weatherald J. Echocardiography-derived stroke volume index is associated with adverse in hospital outcomes in intermediate-risk acute pulmonary embolism: a retrospective cohort study. Chest. 2020;158:1132–42.”

  • We thank the reviewer for these relevant remark and references which have been added to the “bedside echocardiography” section

“As authors pointed out, hemodynamic instability is the main determinant of poor outcome. Clinical criteria often dichotomize patients based on “”hypotension” as defined by SBP<90 mmHg. Recent data have suggested that perhaps SBP is more of a continuous risk measure, rather than a dichotomized variable, although dichotomized variable is easier to apply clinically. Authors should consider mentioning this point and the associated study.

Quezada, D. Jiménez, B. Bikdeli, et al., Systolic blood pressure and mortality in acute symptomatic pulmonary embolism, International Journal of Cardiology, 2019”

  • Thank you for raising this important aspect. As discussed in the “high-risk” section and raised by the reviewer, systolic blood pressure should probably be considered as a linear risk factor rather than a simplistic dichotomised variable. This point has been highlighted in the”high-risk” section

“Other points of clarification: Page 3” In high-risk acute PE, management relies on organ support and prompt reperfusion with thrombolytic therapies.” Consider revising a sentence to include other “”advanced therapeutic options such as percutaneous or surgical embolectomy or mechanical circulatory support” Your sentence may be perceived to suggest that the only option available for high risk PE is thrombolytic therapy.”

  • As suggested by the reviewer, the sentence was revised to include mechanical (percutaneous or surgical) thrombectomy as an alternative to thrombolytic therapy.

“Page 5 “Despite a high sensitivity and negative predictive value, PESI and sPESI lack specificity 159 and their positive predictive value (PPV) for early mortality is low (Table 2).”

The table that you reference describes Sensitivity, specificity, PLR, and NLR, and while they are related to PPV, it is not the same, consider revising the sentence or the table to stay consistent. “

  • We thank the reviewer for this comment. In order to improve consistency between the text and table 2, the mention of positive predictive value has been removed from the text.

“Page 10“7. Improving the safety profile: reduced-dose thrombolytic therapy”

Consider adding percutaneous embolectomy devices which do not use any thrombolytic therapy and therefore may reduce the bleeding risk associated with the thrombolytic use.

Akhilesh K. Sista et al. J Am Coll Cardiol Intv 2021; 14:319-329.

Thomas Tu et al. J Am Coll Cardiol Intv 2019; 12:859-869.”

  • We thank the reviewer for this suggestion. These two relevant references have been added to the paragraph dedicated to catheter-based therapies.

Reviewer #2: “The review manuscript entitled "Risk Stratification in Patients with Acute Pulmonary Embolism: Current evidence and Perspectives” The manuscript is well written.”

  • We thank the reviewer for this supporting comment

“There are several minor points to be addressed: Pls add  the scales and markers in risk stratifiaction;

NEWS-2 score;

FAST score

GDF-15 (ref. - Pol Arch Int Med 2020; 130 (9): 757 – 765;

Copeptin (ref . J Thromb Thrombolysis. 2016; 41: 563-568.

Age-adjusted TnT Eur Respir J. 2015; 45: 1323-1331”

  • Thank you for these relevant references which have been added to the “other scores” section.

Pls add the section about PERT and ref.: Kardiol Pol. 2019; 77: 228-231; Kardiol Pol 2021;79(12):1311-1319

  • Pulmonary embolism response teams may actually have an important role to help defining the best treatment strategy for patients with acute PE, especially for the heterogeneous intermediate-risk group. This aspect, as well as the proposed reference have been added to the conclusion section.

Pls expand the section about CDT (indications, contraindications for CDT); which patients are good candidates for percutaneous intervention;

  • As proposed by the reviewer, the section about CDT has been expanded to mention the indications for percutaneous interventions, as well as the role of circulatory support with ECMO and surgical thrombectomy for high-risk patients with contraindications for systemic thrombolysis. (Section 7. Improving the safety profile: reduced-dose thrombolytic therapy)

Reviewer 2 Report

Review_2022_JCM

The review manuscript entitled "Risk Stratification in Patients with Acute Pulmonary Embolism: Current evidence and Perspectives”

The manuscript is well written.

There are several minor points to be addressed:

Pls add  the scales and markers in risk stratifiaction;

  • NEWS-2 score;
  • FAST score
  • GDF-15 (ref. - Pol Arch Int Med 2020; 130 (9): 757 – 765;
  • Copeptin (ref . J Thromb Thrombolysis. 2016; 41: 563-568.
  • Age-adjusted TnT Eur Respir J. 2015; 45: 1323-1331

Pls add the section about PERT and ref.: Kardiol Pol. 2019; 77: 228-231; Kardiol Pol 2021;79(12):1311-1319

Pls expand the section about CDT (indications, contraindications for CDT); which patients are good candidates for percutaneous intervention;

Author Response

Reviewer #2: “The review manuscript entitled "Risk Stratification in Patients with Acute Pulmonary Embolism: Current evidence and Perspectives” The manuscript is well written.”

  • We thank the reviewer for this supporting comment.

“There are several minor points to be addressed: Pls add  the scales and markers in risk stratifiaction;

NEWS-2 score;

FAST score

GDF-15 (ref. - Pol Arch Int Med 2020; 130 (9): 757 – 765;

Copeptin (ref . J Thromb Thrombolysis. 2016; 41: 563-568.

Age-adjusted TnT Eur Respir J. 2015; 45: 1323-1331”

  • Thank you for these relevant references which have been added to the “other scores” section.

Pls add the section about PERT and ref.: Kardiol Pol. 2019; 77: 228-231; Kardiol Pol 2021;79(12):1311-1319

  • Pulmonary embolism response teams may actually have an important role to help defining the best treatment strategy for patients with acute PE, especially for the heterogeneous intermediate-risk group. This aspect, as well as the proposed reference have been added to the conclusion section.

Pls expand the section about CDT (indications, contraindications for CDT); which patients are good candidates for percutaneous intervention;

  • As proposed by the reviewer, the section about CDT has been expanded to mention the indications for percutaneous interventions, as well as the role of circulatory support with ECMO and surgical thrombectomy for high-risk patients with contraindications for systemic thrombolysis. (Section 7. Improving the safety profile: reduced-dose thrombolytic therapy)
